# The experience of cerebral palsy stigma amongst adults living in the UK and Ireland: A qualitative co-designed project

Kimberley J. Smith[1]*, Jessica Burke[2], Rachel Lawrence[1‡], Emily Oputa[1‡], Ruth Bailey[3]

1 School of Psychology, University of Surrey, Guildford, United Kingdom, 2 School of Physiotherapy, Royal College of Surgeons in Ireland, Dublin, Ireland, 3 School of Social Sciences and Global Studies, The Open University, Milton Keynes, United Kingdom

☯ These authors contributed equally to this work.
‡ RL and EO also contributed equally to this work.
* Kimberley.j.smith@surrey.ac.uk

## Abstract

### Background

There is evidence that cerebral palsy (CP) could be linked to stigma and discrimination, however current evidence is limited to small qualitative studies. The goal of this co-designed survey was to elicit information on experiences of stigma and discrimination amongst a larger sample of adults in the UK and Ireland.

### Methods

Quantitative questions about sources of stigma and qualitative questions designed to elicit information on experiences of stigma were shared via an online survey.

### Results

Eighty-six people completed the qualitative survey, and 5 themes were generated that captured experiences of stigma and discrimination. Theme 1 (rigid stereotypes) captured the lack of awareness about the heterogeneity of CP. Theme 2 (impact on participation) highlighted the difficulties that participants had with participation, particularly in terms of accessibility and sexual relationships. Theme 3 (interpersonal difficulties) included the difficulties people with CP had in interactions with the public such as feel visible in some situations, invisible in others and being infantilised. Theme 4 (systematic discrimination) highlighted discrimination in the workplace, healthcare and broader environment. Theme 5 (negative emotional impact) captured the negative emotional impact that experiences of stigma and discrimination had. Quantitative responses from 48 participants indicated that stigma was a common experience (experienced by 87.5% of respondents), and the most common sources of stigma were the public, classmates and coworkers.

**Data availability statement:** Data cannot be shared publicly because participants did not consent for their data to be shared. The fact that data would not be shared in an open access repository was mentioned in participant information sheet. Participants only consented for data to be shared with the research team. Requests to access anonymised data can be sent to assurance@surrey.ac.uk.

**Funding:** This research was supported by a BA/Leverhulme small research grant (SRG20\200255). The funders had no role in study design, data collection and analysis, decision to publish, or preparation of the manuscript.

**Competing interests:** The authors have declared that no competing interests exist.

## Conclusions

Results indicate that CP is linked to experiences of stigma and discrimination which arise from a lack of understanding of the heterogeneity of CP, a public lack of awareness of how to communicate with people with disabilities, inaccessible environments and negative societal attitudes towards visible impairments. Suggested ways to tackle these issues include improving understanding of CP and removing barriers to accessibility.

## Introduction

Social stigma represents the negative, stereotyped and prejudicial attitudes and beliefs that people within society have towards membership of a particular group [1,2]. Social stigma can also be linked with experiences of discrimination when people are treated differently or denied equality based on being a member of a group or possessing particular characteristics [3]. Experiences of social stigma and discrimination can be harmful to individuals, potentially leading to internalized stigma, poorer quality of life and worsened mental and physical health [3]. Living with a disability has consistently been shown to be associated with notable stigma, discrimination and marginalization [4]. In addition to stigma and discrimination the term 'ablesim' is increasingly used to capture a form of discrimination that favours able-bodied people, resulting in the marginalization of disabled people [5].

The social model of disability highlights that barriers created by society cause impairments to become disabilities [6]. However, the social model of disability has been critiqued for understating the impact that physical impairments have, plus not acknowledging the complex multidimensional factors that contribute to disabling experiences [7,8]. An example of a more integrative framework, the international classification system of functioning, disability and health instead proposes that disability arises from an interaction between impairments with environmental and personal factors which act to limit the ability to take part in activities and participate [9]. Across different models, stigma and discrimination have been identified as environmental factors which can contribute to and exacerbate experiences of disability [6,9].

Cerebral palsy (CP) is the most common lifelong physical disability in the UK, and has been linked with reports of social stigma and discrimination [10]. CP is the diagnostic term used to capture a heterogeneous syndrome that results from an insult or injury to the developing human brain [11]. The most commonly presenting features of CP are difficulties with gross and fine motor functioning, which are linked to issues with gait, movement, balance, posture, muscle spasticity and verbal communication [11,12]. There is a wide spectrum of CP with several sub-types and a breadth of impairment severity measured using the gross motor functioning classification system which ranges from 1 (walks without difficulty in most settings) through to 5 (non-mobile, requiring wheelchair) [11,13]. People with CP can also experience a wide range of co-morbidities such as learning disability, swallowing difficulties, epilepsy, cognitive difficulties, behavioural difficulties and sensory impairments [11,12]. CP

affects everyone differently, but interview studies conducted with people with CP indicate their condition can be stereotyped in negative and potentially harmful ways by both the public and healthcare professionals [14,15]. For instance, adults with CP feel misjudged as having reduced capacity when there are difficulties with speech, or when augmented communication devices are used [14,15]. This can lead to reports of condescension when adults with CP are perceived to be less capable than they are [15,16].

Beyond harmful stereotypes, there is also evidence that people with CP experience social stigma and discrimination related to social participation and accessibility. This is particularly concerning when evidence indicates that wellbeing in people with CP is closely linked with having meaningful social participation and good quality relationships [16–18]. Some people with CP report that they experience issues with developing social and romantic relationships feeling as though people avoid them due to their disability [15] and sometimes feeling less sexually desirable [19]. Some people also report that they are viewed as less employable, with some interview studies revealing worrying accounts of people being refused jobs due to their use of mobility aids [17], or being treated as less capable in a workplace because of the use of augmented communication devices [14]. In addition, those people who use mobility aids report being seen as the mobility aid rather than a person [14,18].

One suggested way to tackle stigma is to use a person-centred approach by meaningfully working with the affected population to involve them in research design and implementation [20]. Furthermore, the disability rights movement highlights the importance of involving people with disability in any work that impacts them: *"Nothing about us without us"*. To the best of our knowledge this is the first co-designed survey aimed to understand experiences of stigma and discrimination. The aim of this study was to co-design an online questionnaire with adults with CP that could be shared online to gather data on the experience of stigma in CP.

## Methods

### Participants

Inclusion criteria included being aged 18 or older, having a diagnosis of CP and living in the UK or Ireland. A further access criterion was being a carer or supporter of an eligible participant who was answering on their behalf.

Participants were recruited through the social media channels of charities that support adults with CP in the UK and Ireland. A link to the survey and researcher contact details were shared through these social media channels and participants could complete the survey online via Qualtrics or contact the researchers to request a hard copy of the questionnaire or complete the survey on the phone. All participants opted to complete the survey online. Easy-read versions of all project materials were also created to increase the accessibility of the survey for people with intellectual disability. A total of 86 eligible participants completed the questionnaire between 1st August 2023 and 1st January 2024, two of these respondents were carers or supporters of an adult with CP. Data collection was stopped when it was felt that no new codes were emerging from data.

### Ethical approval

The study gained ethical approval from the University of Surrey FHMS ethics committee (FHMS 22–23 215). Participants provided informed consent by filling in an online consent form and ticking a box to agree they consented to participate in the study.

### Patient and public involvement (PPI) and co-design

Two adults with CP (JB and RB) were co-investigators on the study who were involved in questionnaire design and analysis. Prior to designing the questionnaire KS, JB and RB held a focus group with 7 adults with CP to ask about experiences of stigma and to generate ideas for questionnaire implementation. The discussion generated by the questions was so rich

that the team decided to use the focus group questions in the online survey. Further input from the focus group identified that resilience, use of mobility aids, speech impairments and psychological wellbeing would be important things to ask about in the questionnaire. Based off this discussion RB and JB suggested we collect quantitative data to examine the mediating role of resilience in the relationship between stigma and wellbeing. Unfortunately, only 40 participants completed this part of the questionnaire as most participants opted to complete the easy-read version of the questionnaire which had been simplified and did not include these questions. This meant that data were underpowered to detect any potential effects. Therefore, a team decision was taken to focus only on descriptive and open-text data. Following data analysis 5 adults from the advisory group were also involved in checking the credibility of interpretation.

### Design

A cross-sectional questionnaire design was used, and the questionnaire was hosted online through Qualtrics.
**Sources of stigma**: Participants were asked a question about whether they had experienced stigma in relation to their CP and a question on how often they had experienced stigma/discrimination in relation to their CP from the following sources: public, classmates, coworkers, healthcare professionals, employer, media, authority figures and family. Responses of once in life, more than once and multiple times were coded as the person having experienced stigma from that source.
**Open-text questions on stigma**: Participants were asked four questions to elicit information on stigma which asked about myths/misconceptions associated with CP, what about CP makes others uncomfortable/embarrassed, whether they believe people with CP are treated differently and any experiences of stigma.

### Data analysis

Quantitative data were analysed by calculating frequencies. Qualitative data were analysed using thematic analysis [21]. The epistemological stance taken for this analysis was that of social constructivism, which holds that how we think and what we know is shaped through our social interactions and our interpretation of these interactions [22]. This approach was taken as stigma is socially constructed, and it is through their interactions with others that people with CP would build their understanding of stigma and discrimination. The lead author (KS) familiarised herself with data by reading and re-reading. She then undertook an initial coding of the data which she then shared with co-authors JB and RB. KS and JB then met to discuss the codes and emergent themes. They then built up a thematic map and themes based on this discussion which was refined after further consultation with RB. These discussions highlighted the importance of participation in narratives and possible alternative interpretations of quotes. This led the lead author to re-work the interpretation further to have a more semantic rather than latent interpretation of participant quotes. This was felt to be important as collecting data online means that the nuances of in-person interaction is lost and latent interpretation would be overly influenced by author interpretation. As a final step, the credibility of results was discussed with an advisory group of 4 adults with CP who found results to be credible and had suggestions to improve the clarity of quantitative findings.

The positionality of the lead author as a non-disabled, white, middle-class and cis-female researcher who has academic expertise in psychology influenced the codes and themes that were generated.

## Results

### Participants

Of the 86 people who completed the online survey the majority (40.2%) were aged 18–34, were female (76.5%), white (87.2%), single (53.5%), had university-level education (57.9%) and were working (52.9%) (see Table 1 for participant characteristics). The majority reported having spastic CP (78.8%), most used some form of mobility aid with 40.4% using wheelchairs and 41.6% reporting using another mobility aid. Finally, most had no speech impairment (65.4%) (see Table 1).

**Table 1. Participant demographics.**

| Category (n participants who answered question) | | N | % |
|---|---|---|---|
| **Age (n=82)** | 18 to 34 | 33 | 40.2% |
| | 35 to 49 | 28 | 34.2% |
| | 50 + | 21 | 25.6% |
| **Gender (n=81)** | Female | 62 | 76.5% |
| | Male | 18 | 22.2% |
| | Other | 1 | 1.2% |
| **Ethnicity (n=86)** | White | 75 | 87.2% |
| | Black | 7 | 8.1% |
| | Other | 4 | 4.6% |
| **Marital status (n=86)** | Married/partnership | 36 | 41.9% |
| | Single | 46 | 53.5% |
| | Widowed or divorced | 4 | 4.6% |
| **Education (n=76)** | High school | 13 | 17.1% |
| | College | 19 | 25.0% |
| | University | 44 | 57.9% |
| **Employment (n=85)** | Unemployed | 18 | 21.1% |
| | Unemployed – long-term sick leave | 13 | 15.3% |
| | Working full/part time | 45 | 52.9% |
| | Retired | 6 | 7.1% |
| | Other | 3 | 3.5% |
| **Subtype CP (n=80)** | Spastic | 63 | 78.8% |
| | Ataxic | 2 | 2.5% |
| | Dystonic | 4 | 5.0% |
| | Mixed | 6 | 7.5% |
| | Other | 5 | 6.3% |
| **Mobility aids (n=84)** | Crutches/stick/walker | 27 | 32.1% |
| | Wheelchair | 34 | 40.4% |
| | Other | 8 | 9.5% |
| | None | 15 | 17.9% |
| **Speech impairment (n=81)** | Communication aids | 5 | 6.2% |
| | Verbal – people can have difficulty understanding | 23 | 28.4% |
| | None | 53 | 65.4% |

Table 1 legend: Responses do not always add up to 86 as participants were always given the option not to respond to a given question for ethical reasons.

## Stigma questions: Quantitative

Of 48 people who answered the quantitative questions a total of 87.5% agreed that they had experienced stigma (62.5% fully agree and 25% mainly agree). Participants (n ranging from 41 to 47) also answered questions on the sources of stigmatisation or discrimination. The four most frequently reported sources of multiple instances of stigma were the public (80%), classmates (57.8%), healthcare professionals (52.2%) and employers (46.3%). See Table 2 for breakdown of responses.

## Stigma questions: Qualitative

A total of 86 people completed the open-text questions which were subsequently analysed using thematic analysis. Through this analysis we generated five inter-related themes (see Fig 1: thematic map) which we firstly describe in turn

**Table 2. Frequency of experiencing stigma and discrimination from different sources.**

| Source of stigma (n participants who answered question) | Multiple times. N (%) | More than once. N (%) | Once. N (%) | Never. N (%) |
|---|---|---|---|---|
| Family (n=47) | 13 (27.7%) | 12 (25.5%) | 3 (6.4%) | 19 (40.4%) |
| Friends (n=46) | 11 (23.9%) | 16 (34.8%) | 6 (13.0%) | 13 (28.3%) |
| Healthcare professionals (n=46) | 24 (52.2%) | 6 (13.0%) | 4 (9.0%) | 12 (26.1%) |
| Co-workers (n=43) | 14 (32.6%) | 21 (48.8%) | 4 (9.3%) | 4 (9.3%) |
| Classmates (n=45) | 26 (57.8%) | 13 (28.9%) | 2 (4.4%) | 4 (8.9%) |
| Employers (n=41) | 19 (46.3%) | 10 (24.3%) | 8 (19.5%) | 4 (9.8%) |
| Public (n=45) | 36 (80%) | 5 (11.1%) | 1 (2.2%) | 3 (6.7%) |
| Authority figures (n=45) | 14 (31.1%) | 11 (24.4%) | 4 (8.9%) | 16 (35.6%) |
| Media (n=44) | 14 (31.8%) | 12 (27.3%) | 4 (9.1%) | 14 (31.8%) |

Table 2 legend: Where numbers do not add up to 48 this is because people opted not to answer that question as people were always given the option not to respond to a question for ethical reasons.

before moving into the relationships between these themes. Theme 1 was titled 'rigid stereotypes' and captured the broad misunderstandings and incorrect assumptions that members of the public hold about CP. The subthemes of theme 1 highlighted the salience of the stereotype that CP was a learning disability and the effort that people with CP went to in disproving stereotypes. Theme 2 was titled 'impact on participation' and described how stigma and discrimination led to issues with participation. Subthemes of theme 2 highlighted barriers to accessibility and the belief that people with CP cannot have families or sex. Theme 3 was titled 'interpersonal difficulties' and demonstrated how stigma and discrimination led to difficult social interactions for people with CP. The subthemes highlighted issues with feeling both visible and invisible as a person with a disability and also the issues with being infantalised and pitied. Theme 4 was titled 'systemic discrimination' and in this theme participants described instances of systemic discrimination in the workplace, healthcare and broader environment. Finally, theme 5 was titled 'negative emotional impact' and this theme highlighted the negative emotional impact that instances of stigma and discrimination could have on people with CP.

Our thematic map (Fig 1) demonstrates the way that these themes interact with one another. 'Rigid stereotypes' (theme 1) interacted with barriers to accessibility to influence participation (theme 2). 'Rigid stereotypes' (theme 1) also influenced how members of the public interacted with people with CP, and could lead to problematic social interactions (theme 3) when people felt they were treated as less capable or ignored because of beliefs about their disability. 'Systemic discrimination' (theme 4) also exacerbated issues with participation (theme 2) and was linked to the difficulties participants had with interpersonal interactions (theme 3). Finally, all the issues described in themes 1–4 could have a negative emotional impact on people with CP (theme 5).

### Theme 1: Rigid stereotypes

For many participants there was a narrative that stigma was linked to the public holding rigid stereotypes about CP, such as beliefs that all people with CP have a learning disability, use wheelchairs and have speech impairments:

> "People believe that having CP means you are a wheelchair user and there is a level of learning disability. They are unaware of the spectrum of CP" (F, 40-49, no mobility aids).

> "That everyone is a wheelchair user, can't speak and can't walk" (F, 30-39, mobility aids)

An additional stereotype many participants shared was the public belief that they are not able to function independently: "That I'm incapable of most things and need help with everything. That I cannot do sports. That I'm incapable of being independent" (F, 30–39, mobility aids).

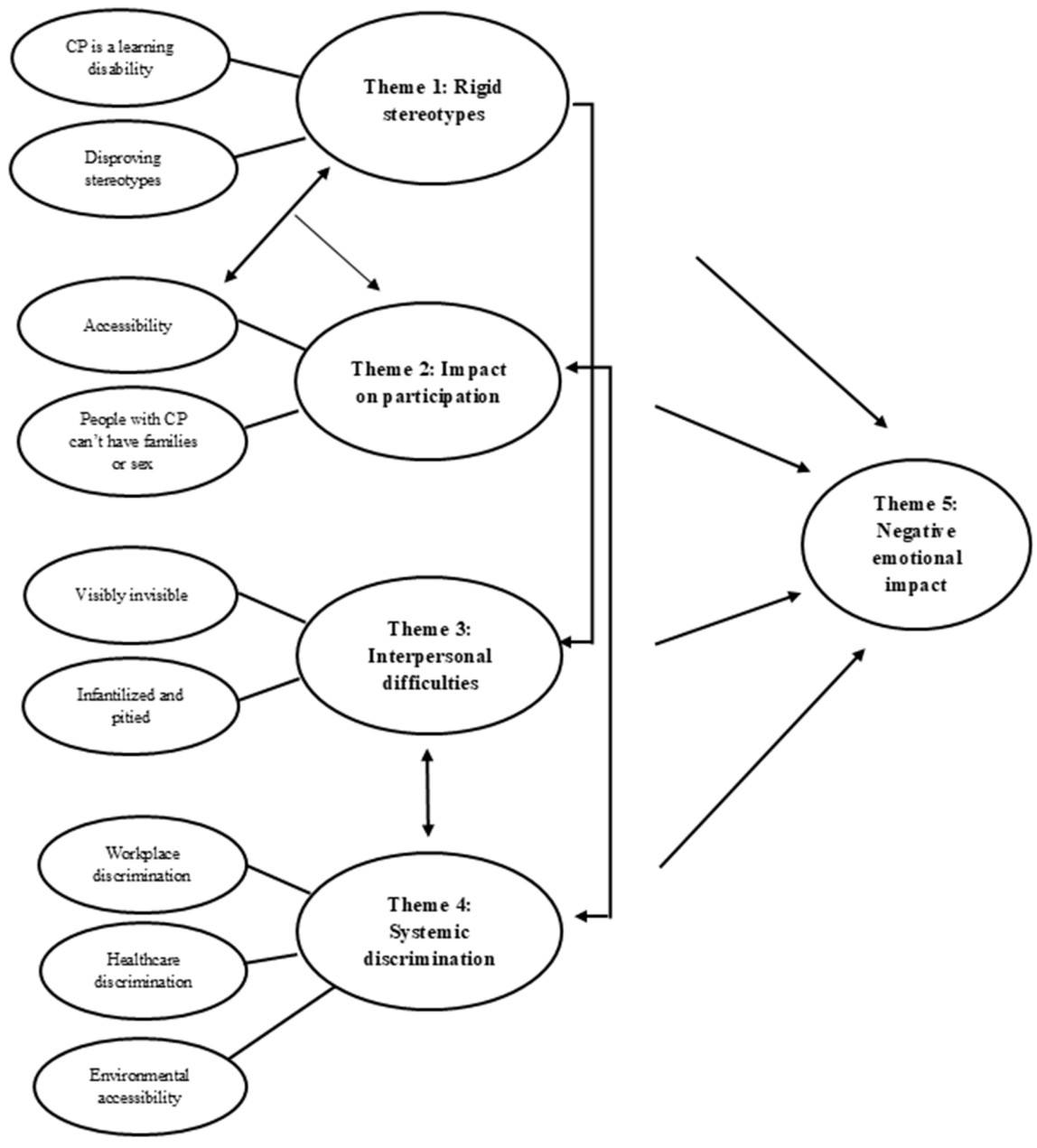

**Fig 1. Thematic map.**

### Sub-theme: CP is a learning disability

The most expressed stereotype was that CP was commonly equated with a learning disability: *"all people with CP have learning disabilities"* (F, 30–39, wheelchair user). The assumption that all people with CP have a learning disability meant that participants frequently felt they were spoken down to: *"people think you are stupid, have low IQ and speak to you like a toddler, people think they have to make decisions for you"* (F, 50–59, wheelchair user) or treated as if they aren't able to understand: *"I've had people speak very slowly. As if I didn't understand"* (F, 50–59, mobility aids).

For many people it was their symptoms of CP that were equated to the assumption they had a learning disability. This was particularly salient for those people who had a speech impairment or visible mobility impairments who often stated that their physical impairment led people to assume they were unable to process information:

*"Because of the way I talk people think I have a hard time understanding things"* (F, 20-29, no mobility aids)

*"the general consensus seems to be if you have something wrong with your body and you are not perfect you probably have something wrong with your mind too"* (F, 50-59, no mobility aids).

**Sub-theme: Disproving stereotypes**

A common narrative that accompanied these stereotypes was the fact that people with CP felt that they had to disprove these rigid stereotypes. This included proving that they were capable of participating: *"Sometimes people assume that you can't do things and are shocked when you prove them wrong"* (F, 50–59, no mobility aids) or that they did not have intellectual impairments: *"I still have to prove I can hold a normal conversation and have mental capability!"* (F, 60–69, wheelchair user). Conversely, many participants also shared that they had to prove that they had CP to others. This could be because they did not fit with the public stereotyped understanding of CP: *"I have found that able bodied people apply their experience of a person with CP to all people with CP. For example and especially at weddings, I've been told that I don't have CP because I'm not in a wheelchair or because my speech is not impaired"* (M, 50–57, no mobility aids) or did not fit broader stereotypes of what a disabled person looks like: *"In the past when I have used a disabled parking bay I get questioned as to why am I parked in this disabled bay since I don't look disabled"* (F, 30–39, mobility aids). This issue of having to prove that they were entitled to use disabled parking spaces was an experience that many participants shared.

**Theme 2: Impact on participation**

The stereotyped lack of understanding that people had about CP (theme 1) directly intersected with people with CP feeling as though assumptions made about their ability to participate: *"There is usually a tendency to underestimate the capabilities of individuals with cerebral palsy, assuming they may need constant assistance or cannot participate fully in various activities"* (M, 20–29, wheelchair user). These assumptions were frequently equated to people with CP feeling as though they were under-estimated because of their impairments (both real and incorrectly stereotyped). It was a source of frustration for people with CP when assumptions were made about their ability to participate: *"I don't like it when people assume I can't do things so don't get included and I have lost friends over this. Examples include golf (I can't play golf), karting (I can drive a kart and have the trophies). My point is I'd like to have the opportunity to decline or make my own judgement call"* (M, 50–59, mobility aids).

The issues with assuming people could not participate due to stereotyping, could lead directly to being given fewer opportunities: *"because there is assumptions that individuals with cerebral palsy also have intellectual disabilities. This misconception affect how some people communicate with us and the opportunities we are given"* (F, 20–27, mobility aids).

**Sub-theme: Accessibility**

Alongside the impact of stereotyping on participation, there was an acknowledgement that issues with accessibility could also impact on participation. Many participants spoke of the need to make reasonable adjustments to facilitate opportunities for participation: *"society should make room for people with this condition to be accepted in employment, healthcare and education"* (M, 20–29, wheelchair user). Not making reasonable adjustments to facilitate participation was linked for some people to feelings of marginalisation: *"There seems to be a sense in which we are sometimes denied the support we need, based on the idea that our lives are viewed as tragic and our own individual issue, rather than as an integral part*

*of society. Frequently, I feel marginalised and othered, as if other people don't want to make the effort to accommodate me, when all I am asking to do is participate in society"* (F, 30–39, wheelchair user)*.*

### Sub-theme: People with CP can't have families or sex

A specific example of assumptions around participation that had particular salience was the belief that people with CP not being able to have sexual relationships or families. Participants shared that they felt people with CP were perceived as asexual *"we are not sexual beings"* (M, 40–49, wheelchair user) and less attractive to others: *"you are undesirable"* (F, 40–49, wheelchair user). Participants also shared accounts of how difficult they had found dating: *"I'm now married, but dating was also awful – I'd do online dating and tell a man I had a disability and 50% of those I told just wouldn't reply"* (F, mobility aids). Some participants described instances where there was an assumption that *"we are incapable of becoming parents"* (F, wheelchair user). Where participants were parents, they recounted difficult interactions where others would question their ability to be a parent: *"A health care professional actually asked if I had someone living with me to help me take care of my unborn child. Though I already had a child which I had brought up on my own before meeting my current partner"* (F, 40–49, mobility aids). In one instance a participant shared that they had experienced abuse due to being a parent with a disability: *"When out with my kids, getting horrible abuse all because I'm a Mum in a wheelchair"* (F, wheelchair user).

### Theme 3: Interpersonal difficulties

Participants spoke about how they could have difficulties in their interactions with members of the public which was felt to be linked to the public being unsure about how to interact with people with CP: *"as CP is deemed a disability, people with it will always be treated differently. People tend to be scared of things they don't know or understand"* (F, 30–39, mobility aids). *"I think people feel embarrassed as they don't know how to ask about it, in general or how to ask about it, even if they are being kind"* (M, 30–39, mobility aids). For some people with CP they felt that the interpersonal difficulties they had could arise from the public being worried about upsetting them: *"Fear of offending me, this fear can lead to discomfort in my social interaction"* (M, 20–29, wheelchair user).

Those participants with speech impairments also shared specific difficulties that arose when they felt arose when others did not understand them: *"Due to having a speech impediment I am never sure does people fully understand me or just nod and agree because it's easier for them and not as embarrassing for them to say sorry I don't (understand) you"* (M, 40–49, mobility aids).

Another common difficulty in social interactions shared by participants was the fact that members of the public often felt entitled to ask personal questions: *"people who deliver my groceries, or work at the bank think it's appropriate to ask intrusive questions about my disability and medical history just because they notice I'm disabled. I'm a busy person, I don't have time to explain hypoxia at birth to everyone I meet"* (F, 20–29, no mobility aids). It was acknowledged that these intrusive questions were not things that would be asked of non-disabled people: *"I think people ask people with cerebral palsy questions that are too personal that they wouldn't ask an able-bodied person"* (F, 30–39, wheelchair user).

Many participants shared that they would prefer members of the public to speak to them directly so that they could tell them about their disability: *"people regularly stare or ask their children to be quiet if they ask why I am walking in a funny way. I'd prefer to explain to the children in an appropriate age-related manner that some people are different and that is okay"*(F, 30–39). However, it was also felt that there was an appropriate way for people to speak with you about your disability: *"I am not even that disabled and I often forget I walk 'funny' so it's always upsetting when people point it out, especially if they point it out to others in front of me (this has happened at work)"* (F, 40–49, no mobility aids).

 

## Sub-theme: Visibly invisible

This subtheme captures the intersecting experiences of being both visible and yet invisible when living with CP. Many participants shared narratives that demonstrated how they would be stared at for being visibly different but would simultaneously be ignored and spoken over as if they were invisible: *"They stare and sometimes ask 'what's wrong with you?' or speak very slowly to me or ignore me and speak to others about me even in my presence"* (F, 40–49, wheelchair user). The visibility of CP led to reactions that ranged from people staring: *"people seem to think it is okay to stare"* (F, 50–59, no mobility aids), to experiencing abuse: *"I've been spat at and laughed at"* (F, 50–59).

Many participants also shared narratives around feeling invisible when people would act as though they were not there: *"I find people will talk to the person you are with rather than you"* (F, 40–49, mobility aids). This issue was frequently mentioned by participants and occurred in a range of settings such as healthcare *"had professionals speak to my husband and not me about me in my presence"* (F, 40–49, wheelchair user) and shops *"The staff in the shop will talk to who I am with regarding a purchase and not to me who has enquired about the purchase in the first place…one time the store attendant gave the person I was with the change and not me"* (F, 60–69, wheelchair user). Participants felt as though they were being avoided because others did not know how to speak with a person with a disability: *"They would rather avoid us than try and understand us"* (M, 50–59, no mobility aids). This feeling of being avoided could be particularly salient for those people who use wheelchairs: *"other people in wheelchairs are seen as…more noticeable as seen as something to be avoided because you look less normal, we are seen as needing to be cared for"* (F, 50–59, wheelchair user).

## Sub-theme: Infantilised and pitied

Participants also spoke about interpersonal difficulties in the ways that they were infantilised and patronised by the public: *"People may treat you like a child. They may make assumptions and try to stop you from following your dreams…people have patted my hand and pitied me. People have spoken patronisingly to me or like I'm a baby. They seem to forget or not understand that I have feelings and can ask questions or challenge them"* (F, 30–39, wheelchair user). This was a notable source of frustration for many people with CP: *"the amount of people who cock their heads and tell me I'm amazing ("awww") makes me roll my eyes so hard"* (F, 20–29, no mobility aids). Alongside been spoken down to, some people also shared how they would be told they are special simply for existing with a disability: *"I got spoken about to my friends and called a real hero"* (M, 30–39, mobility aids). For people with CP, they felt that these reactions came from a societal perception that living with a disability was something negative:

> *"Sometimes I receive pity or sympathy from others who perceive my condition as inherently negative"* (F, 20-29, wheelchair user).

> *"our lives are viewed as tragic and our own individual issue rather than us being an integral part of society"* (F, 30-39, wheelchair user).

## Theme 4: Systemic discrimination

Participants felt that they experienced discrimination in a range of settings which was closely linked to the issues they experienced with participation, namely those issues that arose from accessibility (theme 2) mapping closely onto discriminatory barriers in the environment. Issues with systemic discrimination were also closely linked to interpersonal difficulties (theme 3), particularly with healthcare professionals.

There was a feeling that disability discrimination was not given the same policy priority as the discrimination faced by other minoritised groups: *"Most other groups facing discrimination have high profile events and/or campaigns (e.g., Pride, black lives matter) – disability doesn't have the same profile, nor the support from government policy to remove discrimination and barriers to work and social integration"* (F, mobility aids). This lack of government support could further exacerbate the discrimination faced by adults with CP

The most salient form of discrimination that was discussed by participants was workplace discrimination, with instances of healthcare discrimination and environmental barriers also mentioned.

### Sub-theme: Workplace discrimination

A number of participants shared narratives that they had difficulty gaining a job: *"when applying for a job people often see disability as a barrier to me working for them"* (F, 50–59, mobility aids). This issue was seen as something that was applicable to people living with a disability, rather than CP specifically, and for many people this issue was linked to a lack of awareness or willingness to make the workplace more accessible: *"there is a barrier to employment, through lack of understanding of how to support someone with CP. We can still work there, there just needs to be reasonable adjustments made to our role"* (F, 40–49, no mobility aids). At the most extreme some people felt that despite equality laws that people with disabilities such as CP were still being discriminated against in the workplace:

> *"things like jobs, although you shouldn't and CAN'T be ruled out of jobs (because of) disability, it happens and it's very hard to prove"* (M, 30-39, mobility aids).

> *"In employment – not been given a job and being told it's because of my CP"* (F, wheelchair user).

Alongside experiences of discrimination obtaining employment, participants also provided insight into additional discrimination that could be faced when employed: *"I was introduced to one of the managers. I was told people like you should be using services not providing them. Other workers then looked at me like I was less capable, I always felt undermined like I was tolerated"* (F, 50–59, wheelchair user). It was acknowledged that legally discrimination should not occur, but the discrepancy between workplace policy and workplace reality was problematic for some participants: *"I work in [REDACTED FOR ANONYMITY] …has one of the worst track records of disability discrimination…even though they've got some wonderful anti-discrimination policies, the reality is very different and the senior managers don't care"* (M, 50–59, mobility aids). Other participants acknowledged the difficulties that they had experienced in making their workplace accessible, and in many cases had to advocate for themselves: *"I also have felt disadvantaged in the workplace sometimes because the employer doesn't understand how to modify things to support…I was left to organise it myself"* (F, 50–59, no mobility aids). At the most extreme, when adjustments were not made, this could lead to people leaving employment: *"I had to leave (my) job because they couldn't do enough adaptation for me"* (F, 20–29, mobility aids).

### Sub-theme: Healthcare discrimination

Another common source of systemic discrimination was felt to occur within healthcare settings. For some people this was linked to the theme of interpersonal difficulties, particularly being treated as though they were invisible: *"In medical situations – professional people refusing to talk directly to me"* (F, wheelchair user). For other participants, they felt that this discrimination could extend to treatment due to a lack of understanding of CP from healthcare professionals: *"I think I'm often treated differently when it comes to medical situations, many doctors would rather blame my cerebral palsy diagnosis rather than take the time to treat me properly"* (F, 20–29, wheelchair user). There was also felt to be a lack of specialised healthcare for adults with CP: *"we don't have enough trained medical professionals in older CP"* (F, 40–49, mobility aids). This could be viewed by some people as a form of healthcare discrimination.

### Sub-theme: Environmental barriers

Finally, a number of participants shared accounts of discrimination in the wider environment, most notably with transport and non-accessible buildings which were particularly problematic for wheelchair users: *"refusing me access to transport, places, nightclubs because of my CP"* (F, wheelchair user); *"Physical access to buildings is a major issue for wheelchair*

*users"* (F, 40–49, no mobility aids). It was acknowledged that these barriers would require a major overhaul of the environment: *"Disability is the only protected characteristic that requires changes to the physical environment. In most cases it is easy to see the problem but finding the solution is hard"* (M, 50–59, no mobility aids). However, some participants felt that finding a solution to these barriers was something that society did not feel was important: *"I think the differences in movement that CP causes are seen as a problem in a society that does not prioritise accessible spaces"* (F, 30–39, wheelchair user).

### Theme 5: Negative emotional impact

The experiences of stigma and discrimination captured in previous themes could take an emotional toll on people with CP: *"I experienced negative stereotyping about my disability, this affected my self-confidence greatly"*. (M, 20–29, wheelchair user). Experiences of stigma and discrimination could be linked to feelings of trauma:*"I feel traumatized and often stigmatised in public spaces"* (M, 20–29, wheelchair user). The narratives from participants also highlighted the long-lasting impact that a single experience of stigma or discrimination could have on a person's wellbeing: *"When I was in university some teachers/lecturers said I would not pass my exams before I sat any exams. I did pass all my exams in college. It still has an impact on my self-confidence even today"* (F, 50–59, mobility aids). This quote also demonstrates how people could internalise the stigma that they experienced. Alongside low self-confidence people also mentioned feelings of depression and anxiety resulting from stigma: *"I didn't sleep well for several days, and it made some physical health problems worse. I was distressed and anxious"* (F, 30–39, wheelchair user).

## Discussion

Results from this study highlight that people with CP feel that they experience social stigma from a wide range of sources and that this stigma has a notable impact on the day-to-day lives of people with CP. Quantitative results highlighted that stigma was a common experience and that the public were a particularly frequent source of multiple instances of stigma. Qualitative analysis of open-text responses to questions indicate that participants felt that there was a stereotyped lack of understanding of CP have an adverse effect on the ability to participate and interpersonal interactions. Participants also shared accounts of discrimination and the negative emotional impact that these experiences had on their wellbeing. Due to the issues with the small number of participants who answered the quantitative questions (plus the additional bias introduced by the sampling method), below we focus on the qualitative findings.

### Rigid stereotypes

The public holding rigid stereotypes about all people with CP having the same intellectual, physical and speech impairments was a common experience of stigma in this study, in line with previous research [15,23]. The most salient stereotype in this study was the public belief that CP is equivalent to a learning disability. Like many stereotypes there is some truth in this belief, it is estimated that around half of people with CP have a learning disability [24]. Many people in this study reported that they felt spoken down to because of others assuming that they had a learning disability. This echoes findings from previous research [23,25,26] and reflects a broader societal narrative where the abilities of people with learning disabilities are often underestimated [27]. In tackling this stereotype, it is important to make people aware that some (but not all) people with CP have a learning disability, whilst also tackling the beliefs that are held about people with learning disabilities.

Existing research suggests that there may not be much public awareness of the nuances and intricacies of CP [28]. CP affects people along a spectrum [11,13] yet there is a very unidimensional belief of what CP is. As such, raising awareness of the heterogeneity of CP could be an important way to tackle these rigid and pervasive stereotypes. This is something that many adults with CP do on an individual level as they often feel they need to educate people about what CP is

and prove what they are capable of. For some people educating others about CP could be a way of positively reframing an otherwise negative experience. However, previous research highlights the 'hidden labour' that is involved in managing social interactions for people with a disability [29] indicating that constantly educating others could be burdensome. Many of the stereotypes encountered by people with CP seem both specific to a misunderstanding of the heterogeneity of CP alongside broad stereotypes often applied to people with visible disabilities (e.g., stereotypes that people with disability are asexual, require support and are less intelligent [30,31]. Raising awareness of CP on a population-level (e.g., via media) will be an important way to educate members of the public about what CP is without placing all the onus on people with CP to educate others.

## Impact on participation

Participants highlighted how the lack of public understanding and stereotypes held about CP had a direct influence on participation. Previous research highlights the importance of being able to participate in society for psychological wellbeing, identity formation and a positive self-concept amongst people with CP [17,32]. There is a wealth of evidence that adults with CP experience barriers to participating in employment, independent living, higher education, serious romantic relationships or physical activity [33,34]. This is despite the United Nations convention on the rights of people with disabilities highlighting that people with disabilities have the right to equal participation in political, public and cultural life [35]. This work, coupled with existing research, highlights that many people with CP feel that attitudinal barriers such as stigma and discrimination interact with environmental barriers to prohibit them from equal participation [36].

The difficulties that people have with romantic and sexual participation along with perceptions that people with CP are less attractive and asexual have also been highlighted in previous research amongst people with CP [32,37,38] and is a stereotype applied to people with disability more broadly [30,31]. Previous research suggests that internalising these perceptions can lead to people with CP being less likely to seek romantic relationships and lead to feelings of worthlessness [32]. Research also indicates that the desire for romantic relationships is hugely important to people with CP, but when they feel that others are dismissive of this that it can be a source of notable distress [26]. This indicates that being sensitive to the romantic and sexual needs of people with CP is important for their wellbeing.

## Interpersonal difficulties

Participants shared the stigmatising interpersonal difficulties that they had with others. Broad difficulties were reported whereby people with CP felt that others did not understand how to speak to them as a disabled person. Research from the UK charity SCOPE highlights that 67% of the public feel uncomfortable around people with disability [39]. This was linked to issues such as worrying about saying the wrong thing or offending, not wanting to patronise, not understanding disability and a lack of experience of being around people with disability [39]. Intrusive questioning was also mentioned by participants. This has been linked to an ableist privilege whereby non-disabled people feel they have the right to ask overly personal questions to people with a visible disability so that they can 'diagnose' someone whose body they perceive as different [40,41].

In addition, people with CP also reported feeling both very visible (stared at) while also being treated as if they were invisible (being ignored and spoken over). These seemingly contradictory experiences of visibility and invisibility are commonly reported by people with CP [26,36] and people with different visible disabilities [31,40,42]. Feelings related to visibility and invisibility were more salient for those people in wheelchairs (who are possibly more readily identifiable as having an impairment), a finding consistent with other studies in wheelchair users [43]. These issues have again been linked to an ableist privilege where non-disabled people feel that they have a right to stare at bodies that they perceive as different [40,44]. Furthermore, researchers have linked the ignoring of those with disabilities to the public feeling uncomfortable and unsure about how to interact with people with a disability [31].

 

Many participants also shared stories on how members of the public could infantilize and pity them. Infantilization and patronization are common experiences amongst people with a range of visible disabilities, including CP [26,31,43]. The paternalistic attitudes and infantilization people with a disability experience is something that links into a broader ableist narrative that people with impairments are victims [4] and stereotypes that people with disability are less competent [45,46] and as such require support. These beliefs lead to experiences of 'benevolent ableism' where people with disability inspire praise for completing everyday activities, pity for having a disability and paternalistic unwanted offers of help [5].

These multiple difficult interactions can leave people with a disability feeling de-humanised [42]. Furthermore, research highlights that the burden of managing these problematic social interactions tend to fall on the person with a disability [29].

### Systemic discrimination

Participants in this study reported a range of experiences of discrimination in employment and healthcare settings alongside sharing their concerns about environmental barriers. The most salient sub-theme was on workplace discrimination, both in obtaining employment and then the discrimination that occurred when in employment. The discrimination that participants reported experienced in gaining employment aligns with previous qualitative research amongst adults with CP [14,17,38]. Existing evidence suggests that people with CP worry about being treated equally when applying for jobs [47] or even don't apply for jobs because of fears of stigmatisation [32]. When in the workplace there were additional reports of discrimination in terms of being treated as less competent or difficulties with gaining reasonable adjustments to their workplace. People with CP acknowledge that attitudes towards people with a disability can be a significant barrier to employment [14]. Furthermore, existing evidence suggests that managers don't always feel confident in employing people with disabilities and can have concerns about things such as legal liability and cost [48]. However, positive attitudes towards people with a disability, physically accessible buildings and reasonable adjustments are all linked with a more positive workplace experience as well as ways to retain people with disability in the workforce [17,32]. This suggests that removing barriers and challenging ableist attitudes could be important ways to facilitate the employment of people with CP.

Participants also mentioned discrimination within the healthcare environment. Previous research has highlighted that people with CP feel as though their condition is not understood by healthcare professionals [15]. This is reinforced by research in medical students showing that they had negative attitudes and a poor understanding of CP [49]. Furthermore, in line with our work, people with CP can report difficult interactions with healthcare staff such as being patronized or ignored [25]. This suggests that educating healthcare staff about CP and how to effectively communicate with people with CP could help reduce discrimination in this setting.

Linked to the issues with accessibility mentioned in the participation theme, environmental barriers were notable sources of discrimination. Numerous studies emphasise the difficulties that adults with CP face due to environmental barriers [32,36,50] and the need for more physical accessibility [15]. Improving accessibility in the broader environment would be an important way to reduce discrimination.

### Negative emotional impact

Our work also highlighted the negative emotional impact that stigma and discrimination take on people with CP. Previous research has highlighted the negative impact that stigma and discrimination have on mental health, wellbeing and quality of life [3]. Previous research indicates that a high proportion of people with CP have mental health difficulties [51,52] and it is possible that experiences of stigma and discrimination could be contributing to these difficulties. However, it should be acknowledged that amongst people with disability there can be a range of emotional reactions to stigma and discrimination including internalized ableism, indifference, negative affectivity, anger or empowerment [53].

**Strengths and limitations**

The strengths of this study include the large number of participants who provided qualitative data, the involvement of adults with CP throughout the research process and the steps we took to sense check data interpretation. However, we could not carry out the full quantitative analysis as intended due to low power. It is also worth noting that the implications that can be drawn from the quantitative data are limited by the fact that this is not a representative sample to draw implications on the prevalence of stigma from. This is due to the low n, the potentially biased sampling method (this was advertised as a study on stigma) and the types of people that took part in the survey. Most participants self-completed the questionnaire online and had a high level of education. This indicates that the sample was not necessarily representative of the heterogenous group of people with CP. We also do not have data on the GMFCS levels of participants which could have given more nuanced insight into the range of mobility impairments experienced by participants.

It is also important to acknowledge that as stigma is a culturally-bound phenomenon that the experiences of stigma and discrimination discussed in this study may be experienced differently across different cultures. Finally, as this study was explicitly advertised as a study on CP stigma it is worth acknowledging that most people who self-selected to take part would have had a vested interest in this topic and so may have been more likely to have experienced stigma and discrimination.

**Implications: tackling CP stigma and discrimination**

Results from this study suggest that an important way to tackle stigma and discrimination experienced by adults with CP is to improve the understanding of CP, most notably amongst the public, media and healthcare systems. This aligns with work from scholars which suggests that the reduction of social stigma can be achieved by improving the knowledge of the public to reduce misinformation, change attitudes to reduce prejudice and change behaviours to reduce discrimination [54].

Another salient finding from this study which was seen across different themes was that improving accessibility is seen as an important way to facilitate participation and remove these significant discriminatory barriers. The central role that accessibility plays in the lives of people with CP has been discussed in other studies, in particular how a lack of accessibility can prohibit participation [17,36]. Conversely, when people encounter positive attitudes and environmental adaptation these can facilitate participation and a sense of positive wellbeing [38].

There is no single simple solution to tackling CP stigma and discrimination. Livneh et al [55] proposed potential ways to tackle the stigma experienced by people with physical and sensory disabilities could include direct contact, educational interventions, and persuasive communication to change beliefs. However, it is also worth being mindful that previous research on reduction of stigma highlights that any intervention needs to be multi-component working on the different levels that stigma operates at from the micro (internal, interpersonal) through to the macro (community) and meso (structural) [56]. It is also important to continue to involve people with CP in this research through co-designing potential interventions.

**Conclusions**

Results from this project highlight the various forms that stigma and discrimination take in the lives of adults with CP including rigid stereotyping from the public, difficulties with participation, interpersonal difficulties, discrimination from different structures and the negative emotional impact of these experiences. Based off our findings we suggest that improving the understanding of the public about the heterogeneity of CP and removing barriers to accessibility could be core ways to tackle stigma. Our findings also suggest that despite legislation, that discrimination continues to be a pervasive issue for this group. This indicates that more work needs to be done to ensure that these laws are upheld across settings where discrimination continues to occur.

## Acknowledgments

We would also like to thank all members of the advisory group who contributed to this study: Karen Rowe, Sarah Cooper, Jacqueline McTaggart, Katy Evans, Pippa Mundy and Richard Luke.

## Author contributions

**Conceptualization:** Kimberley J Smith, Jessica Burke, Ruth Bailey.

**Data curation:** Kimberley J Smith, Rachel Lawrence.

**Formal analysis:** Kimberley J Smith, Jessica Burke, Ruth Bailey.

**Funding acquisition:** Kimberley J Smith.

**Investigation:** Kimberley J Smith, Rachel Lawrence, Emily Oputa, Ruth Bailey.

**Methodology:** Kimberley J Smith, Jessica Burke, Ruth Bailey.

**Project administration:** Kimberley J Smith, Rachel Lawrence, Emily Oputa.

**Supervision:** Kimberley J Smith.

**Writing – original draft:** Kimberley J Smith.

**Writing – review & editing:** Kimberley J Smith, Jessica Burke, Rachel Lawrence, Emily Oputa, Ruth Bailey.

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
