## [Decision Letter · Decision Letter 0]

30 Jan 2025

Dear Dr. Smith,

Thank you for submitting your manuscript to PLOS ONE. After careful consideration, we feel that it has merit but does not fully meet PLOS ONE’s publication criteria as it currently stands. Therefore, we invite you to submit a revised version of the manuscript that addresses the points raised during the review process.

 Could you please revise the manuscript to carefully address the concerns raised? Please submit your revised manuscript by Mar 16 2025 11:59PM. If you will need more time than this to complete your revisions, please reply to this message or contact the journal office at ?>plosone@plos.org . A rebuttal letter that responds to each point raised by the academic editor and reviewer(s). You should upload this letter as a separate file labeled 'Response to Reviewers'.A marked-up copy of your manuscript that highlights changes made to the original version. You should upload this as a separate file labeled 'Revised Manuscript with Track Changes'.An unmarked version of your revised paper without tracked changes. You should upload this as a separate file labeled 'Manuscript'.

We look forward to receiving your revised manuscript.

Kind regards,

Helen Howard

Staff Editor

PLOS ONE

“This research was supported by a BA/Leverhulme small research grant (SRG20\200255).”

“This research was supported by a BA/Leverhulme small research grant (SRG20\200255). We would also like to thank all members of the advisory group who contributed to this study: Karen Rowe, Sarah Cooper, Jacqueline McTaggart, Katy Evans, Pippa Mundy and Richard Luke.”

“This research was supported by a BA/Leverhulme small research grant (SRG20\200255).”

Reviewers' comments:

Reviewer's Responses to Questions

**Comments to the Author**

1. Is the manuscript technically sound, and do the data support the conclusions?

Reviewer #1: Yes

2. Has the statistical analysis been performed appropriately and rigorously?

Reviewer #1: Yes

3. Have the authors made all data underlying the findings in their manuscript fully available?

Reviewer #1: No

4. Is the manuscript presented in an intelligible fashion and written in standard English?

Reviewer #1: Yes

Reviewer #1: The experience of cerebral palsy stigma amongst adults living in the UK and Ireland: A

qualitative co-designed project

General comments:

The article is interesting and well written. It helps understand the experiences of people with CP, and importantly was co-designed together with people who have CP. It adds to the literature and is a definitive asset to scholarly knowledge as well as providing practical implications.

Abstract:

The results as presented in the abstract are difficult to follow. The themes should all be presented in the same manner (for example theme 2: impact on participation). It is difficult to understand the flow between themes themselves and the complicated flow chart in words. The themes are presented most clearly in the conclusion (though the 5th is missing). Please rewrite the results in a clearer presentation.

Also, there is no mention of the quantitative results. This needs fixing (or removing the quantitative section)

Introduction:

The flow is good and explains the importance.

In the description of the ICF model, the order should be reversed in the sentence “to participate and take part in activities” in order to be in line with the model.

The social model of disability is not mentioned, despite that its message resonates in this article. It should be added to the introduction or the discussion (or both).

The paragraph describing CP says that the most common presenting feature includes fine motor, but gait is gross motor. It is also not correct to say that fine motor function leads to muscle spasticity nor balance nor posture issues. This whole section needs to be written more accurately.

The paragraph about CP also describes the GMFCS, which is very important, but the GMFCS is never mentioned in the article after this. It would have been useful in the demographic description of the participants, but I suppose if it was a self-reported then some people might not know their GMFCS level. If the GMFCS is not used in the research, it isn’t necessary to explain it. The heterogeneity can be described without it.

Paragraph starting “In order to tackle stigma”: the person-centred approach is very important for many reasons. Preventing stigma an important reason, but the approach should not be shrunk only to this. Please re-phrase this.

Methods:

It is not clear how the participants were recruited. Was the survey a link on the social media channels of charities, or was there an add asking people to contact the researchers and then they received a link to the survey or an email? If the survey was in on the site, how could people receive hard copies or phone help? There is no data as to how many of the respondents were people with CP and how many were proxies (care givers). This is very important as if all participants filled out the survey themselves, then the results only reflect this populations point of view.

PPI- please write out abbreviations in full the first time used

The section about resilience and wellbeing can be shortened and the results that only 40 answered should be in the results. This should also be explored briefly in the discussion.

Sources of stigma: It is hard to understand how “once in life” can be put in the same category as “multiple times”.

It would be useful to have the survey in an appendix.

Results:

Table 1:

Each descriptive category has a different N which is vey confusing. They range from 76 (education) to 86 (marital status). If this is the case, then write the N in the first column with the category.

Athetotic is a form of dystonic CP. Hemiplegia is a description of where there is tone (half the body versus diplegia ....) whereas the other descriptions are of the type of tone. Most people with hemiplegia are spastic. Is the “other” mobility ais a scooter? The table needs more work.

In the written description of the demographics most data is presented to one decimal except 40% using wheelchair. It needs to be consistent.

Stigma questions: consider rephasing: Participant (n= 41- 47) reported that the sources of stigmatisation or discrimination was most frequently public.....

Please explain how the data presented in this paragraph was calculated as it doesn’t fit with the data in Table 2. For example, 89.1% (public): 80% + 11.1% doesn't add up to 89.1%. This is the same with other data taken from table 2. The numbers don’t seem to fit the data in the table, so please explain how this was done.

Table 2: please add the N to each row in the first column, then there is no need for the explanation at the end.

Stigma questions: Qualitative:

Please end quotation marks around the phrase rigid stereotypes

Please try to write the paragraph about the five themes in a clearer manner and write all themes in the same method (sometimes the theme itself is in parenthesis, sometimes the number is also and sometimes just the number). It is hard to follow. In the thematic map, the arrow between theme 1 and 2 should be uni-directional. Only 1 to 2 and not back also, if I understood correctly. Also, the sub-themes should be stated clearly (i.e. write subthemes at the top of the column or in each circle).

Rigid Stereotypes:

The sentence “In some cases, this misunderstanding that people with CP are not able to function could lead to unsolicited offers of help which were not always required” does not seem to fit the theme. This is not the stereotype, rather peoples’ actions.

Discussion:

The summery of the themes in the first paragraph is very well done. The clearest in the article.

There doesn’t seem to be any discussion of the quantitative data. Despite that this data is weaker than the qualitative data, it should be discussed or removed from the article. Please state the importance of data found from table 2, both in the discussion and conclusion.

The inclusion of other people with other disabilities or other people in wheelchairs is very interesting.

Limitations:

The fact that the people who responded to the survey were all literate, and many with post-secondary education, should be included in the limitations as there is little representation of the people with CP in the lower end of the spectrum (and we don’t know if any proxies participated to give information for them). The lack of GMFCS is also a limitation. The wide spectrum of people with CP is discussed, but there is little information about the representation of the people with CP who are lower functioning.

Implications:

Please explain how positive impact of seeking support from peers with CP will tackle stigma.

Conclusions:

The 5th theme is missing in the conclusion (which is a pity as it is very interesting).

The sentence “Many of the ways in which stigma and discrimination present for this group are common experiences amongst people with different visible disabilities” does not seem to be a conclusion from the data of this study.

As stated at the start, the article is very interesting, well written and adds to knowledge in the field. The comments are relatively minor and changes will make this article a great asset.

**Do you want your identity to be public for this peer review?** For information about this choice, including consent withdrawal, please see our Privacy Policy

Reviewer #1: No

---

## [Author Response · Author response to Decision Letter 1]

17 Feb 2025

The experience of cerebral palsy stigma amongst adults living in the UK and Ireland: A

qualitative co-designed project

General comments:

The article is interesting and well written. It helps understand the experiences of people with CP, and importantly was co-designed together with people who have CP. It adds to the literature and is a definitive asset to scholarly knowledge as well as providing practical implications.

Abstract:

The results as presented in the abstract are difficult to follow. The themes should all be presented in the same manner (for example theme 2: impact on participation). It is difficult to understand the flow between themes themselves and the complicated flow chart in words. The themes are presented most clearly in the conclusion (though the 5th is missing). Please rewrite the results in a clearer presentation.

Also, there is no mention of the quantitative results. This needs fixing (or removing the quantitative section)

Response: As the comment mentions that the interplay between different themes complicates the clear presentation of results, we instead focus on the themes themselves (and the interplay between these is something that is picked up on in the results where we have the scope to expand on this). We have also included information about the quantitative part of the study (see below):

“Theme 1 (rigid stereotypes) captured the lack of awareness about the heterogeneity of CP. Theme 2 (impact on participation) highlighted the difficulties that participants had with participation, particularly in terms of accessibility and sexual relationships. Theme 3 (interpersonal difficulties) included the difficulties people with CP had in interactions with the public such as feel visible in some situations, invisible in others and being infantilised. Theme 4 (systematic discrimination) highlighted discrimination in the workplace, healthcare and broader environment. Theme 5 (negative emotional impact) captured the negative emotional impact that experiences of stigma and discrimination had. Quantitative responses from 48 participants indicated that stigma was a common experience (experienced by 87.5% of respondents), and the most common sources of stigma were the public, classmates and coworkers.”

Introduction:

The flow is good and explains the importance.

In the description of the ICF model, the order should be reversed in the sentence “to participate and take part in activities” in order to be in line with the model.

Response: This has been amended, please see below:

“The international classification system of functioning, disability and health proposes that disability arises from an interaction between impairments with environmental and personal factors which act to limit the ability to take part in activities and participate (6).”

The social model of disability is not mentioned, despite that its message resonates in this article. It should be added to the introduction or the discussion (or both).

Response: This had been a conscious decision because of the critiques commonly levelled against the social model of disability (and our desire to focus on a more integrated model of disability), though we do concede that it’s omission from our work could be seen as problematic. As such we have included the social model into the introduction, whilst being mindful of the issues with the model (please see re-worked section below):

“The social model of disability highlights that barriers created by society cause impairments to become disabilities (6). However, the social model of disability has been critiqued for understating the impact that physical impairments have, plus not acknowledging the complex multidimensional factors that contribute to disabling experiences (7, 8). An example of a more integrative framework, the international classification system of functioning, disability and health instead proposes that disability arises from an interaction between impairments with environmental and personal factors which act to limit the ability to take part in activities and participate (9). Across different models, stigma and discrimination have been identified as environmental factors which can contribute to and exacerbate experiences of disability (6, 9).”

The paragraph describing CP says that the most common presenting feature includes fine motor, but gait is gross motor. It is also not correct to say that fine motor function leads to muscle spasticity nor balance nor posture issues. This whole section needs to be written more accurately.

Response: Thank you for pointing out this typographical error, this section should have read gross and fine motor functioning and has been re-worked as such:

“The most commonly presenting features of CP are difficulties with gross and fine motor functioning, which are linked to issues with gait, movement, balance, posture, muscle spasticity and verbal communication (11, 12).”

The paragraph about CP also describes the GMFCS, which is very important, but the GMFCS is never mentioned in the article after this. It would have been useful in the demographic description of the participants, but I suppose if it was a self-reported then some people might not know their GMFCS level. If the GMFCS is not used in the research, it isn’t necessary to explain it. The heterogeneity can be described without it.

Response: We do appreciate this point, but feel that leaving the GMFCS levels in is important as in the results we do talk about the stereotypes held around the types of mobility impairments and we feel it is important to highlight that there is a breadth of mobility impairments associated with CP.

Paragraph starting “In order to tackle stigma”: the person-centred approach is very important for many reasons. Preventing stigma an important reason, but the approach should not be shrunk only to this. Please re-phrase this.

Response: We can see your point, and have re-framed the start of the sentence to state that this is one way we can tackle stigma:

“One suggested way to tackle stigma is to use a person-centred approach by meaningfully working with the affected population to involve them in research design and implementation (20).”

Methods:

It is not clear how the participants were recruited. Was the survey a link on the social media channels of charities, or was there an add asking people to contact the researchers and then they received a link to the survey or an email? If the survey was in on the site, how could people receive hard copies or phone help? There is no data as to how many of the respondents were people with CP and how many were proxies (care givers). This is very important as if all participants filled out the survey themselves, then the results only reflect this populations point of view.

Response: Thank you for this feedback. We have included additional information and re-worked this section of the paper so that it is hopefully clearer how people were recruited and who completed the surveys:

“Participants were recruited through the social media channels of charities that support adults with CP in the UK and Ireland. A link to the survey and researcher contact details were shared through these social media channels and participants could complete the survey online via Qualtrics or contact the researchers to request a hard copy of the questionnaire or complete the survey on the phone. All participants opted to complete the survey online. A total of 86 eligible participants completed the questionnaire between 1st August 2023 and 1st January 2024, two of these respondents were carers or supporters of an adult with CP.”

We have also included information on the fact that easy-read project materials were available as this is important information on considerations we had to make recruitment more inclusive that was previously missing:

“Easy-read versions of all project materials were also created to increase the accessibility of the survey for people with intellectual disability.”

PPI- please write out abbreviations in full the first time used

Response: This has been amended, please see below:

“Patient and public involvement (PPI) and co-design”

The section about resilience and wellbeing can be shortened and the results that only 40 answered should be in the results. This should also be explored briefly in the discussion.

Response: We are reluctant to delete this detail as this highlights the collaborative approach that we took to co-design the survey. As such, we have opted to leave this information in. However, we have included additional information in this section for why only 40 people completed the additional questions (this is because most people clicked on the easy-read version of the survey which was substantially simplified and did not include the additional questions). As such, we have opted not to discuss in the discussion as it isn’t needed.

“Unfortunately, only 40 participants completed this part of the questionnaire as most participants opted to complete the easy-read version of the questionnaire which had been simplified and did not include these questions.”

Sources of stigma: It is hard to understand how “once in life” can be put in the same category as “multiple times”.

Response: We have amended this section of the results section so that we only report on people who have experienced stigma from that source multiple times. We have chosen to focus narrative results only on the 4 most reported sources (who had percentages around 50% or more):

“The four most frequently reported sources of multiple instances of stigma were the public (80%), classmates (57.8%), healthcare professionals (52.2%) and employers (46.3%).”

It would be useful to have the survey in an appendix.

Response: As the survey includes other validated scales that it may be problematic to reproduce we opted not to include it as an appendix. However, we feel we have included enough information on the types of questions that were asked in the methods.

Results:

Table 1:

Each descriptive category has a different N which is vey confusing. They range from 76 (education) to 86 (marital status). If this is the case, then write the N in the first column with the category.

Response: We have added total n’s into the first column in brackets as requested. For ethical reasons participants were always given the option not to respond to a given question (as some people prefer not to share personal information). This means not all participants have answered all questions. This is now clarified below the table:

“Responses do not always add up to 86 as participants were always given the option not to respond to a given question for ethical reasons.”

Athetotic is a form of dystonic CP. Hemiplegia is a description of where there is tone (half the body versus diplegia ....) whereas the other descriptions are of the type of tone. Most people with hemiplegia are spastic. Is the “other” mobility ais a scooter? The table needs more work.

Response: We appreciate you highlighting this point, though it is worth noting that this survey was developed with people with CP who did not highlight the use of these subtypes of CP as a concern. However we have chosen to use four main subtypes as identified here (spastic, dyskinetic, ataxic and mixed): https://cerebralpalsy.org.au/cerebral-palsy/types/ and then collapsed the other responses into ‘other’.

In terms of ‘other’ response for mobility, this was a response that participants could choose if they did not use one of the other identified mobility aids. As such, we have left as other as it would be unwise for us to speculate on what this could be.

In the written description of the demographics most data is presented to one decimal except 40% using wheelchair. It needs to be consistent.

Response: Thank you for pointing this out. We have amended the 40% to 40.4%

Stigma questions: consider rephasing: Participant (n= 41- 47) reported that the sources of stigmatisation or discrimination was most frequently public.....

Response: The section reads as this following suggested amendments:

“Participants (n ranging from 41 to 47) also answered questions on the sources of stigmatisation or discrimination. The four most frequently reported sources of multiple instances of stigma were the public (80%), classmates (57.8%), healthcare professionals (52.2%) and employers (46.3%).”

Please explain how the data presented in this paragraph was calculated as it doesn’t fit with the data in Table 2. For example, 89.1% (public): 80% + 11.1% doesn't add up to 89.1%. This is the same with other data taken from table 2. The numbers don’t seem to fit the data in the table, so please explain how this was done.

Response: We have amended this section in line with advice above (so now we only focus on multiple instances of stigma) and so no longer have these numbers.

Table 2: please add the N to each row in the first column, then there is no need for the explanation at the end.

Response: We have amended the table as requested (with n’s in brackets after each category) and added the following description under the table:

“Where numbers do not add up to 48 this is because people opted not to answer that question as people were always given the option not to respond to a question for ethical reasons.”

Stigma questions: Qualitative:

Please end quotation marks around the phrase rigid stereotypes

Response: This has been corrected

Please try to write the paragraph about the five themes in a clearer manner and write all themes in the same method (sometimes the theme itself is in parenthesis, sometimes the number is also and sometimes just the number). It is hard to follow. In the thematic map, the arrow between theme 1 and 2 should be uni-directional. Only 1 to 2 and not back also, if I understood correctly. Also, the sub-themes should be stated clearly (i.e. write subthemes at the top of the column or in each circle).

Response: Thank you for highlighting this to us. We have decided to separate out the description of themes from the interaction between themes. We hope that this makes the section easier to follow. Please see below:

“ Through this analysis we generated five inter-related themes (see figure 1: thematic map) which we firstly describe in turn before moving into the relationships between these themes. Theme 1 was titled ‘rigid stereotypes’ and captured the broad misunderstandings and incorrect assumptions that members of the public hold about CP. The subthemes of theme 1 highlighted the salience of the stereotype that CP was a learning disability and the effort that people with CP went to in disproving stereotypes. Theme 2 was titled ‘impact on participation’ and described how stigma and discrimination led to issues with participation. Subthemes of theme 2 highlighted barriers to accessibility and the belief that people with CP cannot have families or sex. Theme 3 was titled ‘interpersonal difficulties’ and demonstrated how stigma and discrimination led to difficult social interactions for people with CP. The subthemes highlighted issues with feeling both visible and invisible as a person with a disability and also the issues with being infantalised and pitied. Theme 4 was titled ‘systemic discrimination’ and in this theme participants described instances of systemic discrimination in the workplace, healthcare and broader environment. Finally, theme 5 was titled ‘negative emotional impact’ and this theme highlighted the negative emotional impact that instances of stigma and discrimination could have on people with CP.

Our thematic map demonstrates the way that these themes interact with one another. ‘Rigid stereotypes’ (theme 1) interacted with barriers to accessibility to influence participation (theme 2). ‘Rigid stereotypes’ (theme 1) also influenced how members of the public interacted with people with CP, and could lead to problematic social interactions (theme 3) when people felt they were treated as less capable or ignored because of beliefs about their disability. ‘Systemic discrimination’ (theme 4) also exacerbated issues with participation (theme 2) and was linked to the difficulties participants had with interpersonal interactions (theme 3). Finally, all the issues described in themes 1-4 could have a negative emotional impact on p

---

## [Decision Letter · Decision Letter 1]

19 Aug 2025

The experience of cerebral palsy stigma amongst adults living in the UK and Ireland: A qualitative co-designed project

PONE-D-24-53575R1

Dear Dr. Smith,

We’re pleased to inform you that your manuscript has been judged scientifically suitable for publication and will be formally accepted for publication once it meets all outstanding technical requirements.

Kind regards,

Emily Lund

Academic Editor

PLOS ONE

Additional Editor Comments (optional):

I believe that all comments have been adequately addressed.

Reviewers' comments:

Reviewer's Responses to Questions

**Comments to the Author**

Reviewer #1: All comments have been addressed

2. Is the manuscript technically sound, and do the data support the conclusions?

Reviewer #1: Yes

3. Has the statistical analysis been performed appropriately and rigorously?

Reviewer #1: Yes

4. Have the authors made all data underlying the findings in their manuscript fully available?

Reviewer #1: No

5. Is the manuscript presented in an intelligible fashion and written in standard English?

Reviewer #1: Yes

Reviewer #1: The article is excellent, and very interesting. I hope it will be well read and help people understand the experiences of people with CP.

**Do you want your identity to be public for this peer review?** For information about this choice, including consent withdrawal, please see our Privacy Policy

Reviewer #1: No

---

## [Editor Report · Acceptance letter]

PONE-D-24-53575R1

PLOS ONE

Dear Dr. Smith,

I'm pleased to inform you that your manuscript has been deemed suitable for publication in PLOS ONE. Congratulations! Your manuscript is now being handed over to our production team.

Kind regards,

on behalf of

Dr. PLOS Manuscript Reassignment

Staff Editor

PLOS ONE